# OPEN LOOP HYPERPARAMETER OPTIMIZATION AND DETERMINANTAL POINT PROCESSES

## ABSTRACT

Driven by the need for parallelizable hyperparameter optimization methods, this paper studies *open loop* search methods in the sense that the sequence is predetermined and can be generated before a single configuration is evaluated. Examples include grid search, uniform random search, low discrepancy sequences, and other sampling distributions. In particular, we propose the use of $k$-determinantal point processes in hyperparameter optimization via random search. Compared to conventional uniform random search where hyperparameter settings are sampled independently, a $k$-DPP promotes diversity. We describe an approach that transforms hyperparameter search spaces for efficient use with a $k$-DPP. In addition, we introduce a novel Metropolis-Hastings algorithm which can sample from $k$-DPPs defined over spaces with a mixture of discrete and continuous dimensions. Our experiments show significant benefits over uniform random search in realistic scenarios with a limited budget for training supervised learners, whether in serial or parallel.

## 1 INTRODUCTION

Hyperparameter values—regularization strength, model family choices like depth of a neural network or which nonlinear functions to use, procedural elements like dropout rates, stochastic gradient descent step sizes, and data preprocessing choices—can make the difference between a successful application of machine learning and a wasted effort. To search among many hyperparameter values requires repeated execution of often-expensive learning algorithms, creating a major obstacle for practitioners and researchers alike.

In general, on request/iteration $k$, a hyperparameter searcher suggests a hyperparameter configuration $x_k$, a worker trains a model using $x_k$, and returns a validation loss of $y_k$ computed on a hold out set. In this work we say a hyperparameter searcher is **open loop** if $x_k$ depends only on $\{x_i\}_{i=1}^{k-1}$; examples include choosing $x_k$ uniformly at random (Bergstra et al., 2011a), or $x_k$ coming from a low-discrepancy sequence (c.f., Iacò (2015)). We say a searcher is **closed loop** if $x_k$ depends on both the past configurations and validation losses $\{(x_i, y_i)\}_{i=1}^{k-1}$; examples include Bayesian optimization (Snoek et al., 2012) and recent reinforcement learning methods (Zoph & Le, 2016). Note that open loop methods can draw an infinite sequence of configurations before training a single model, whereas closed loop methods rely on validation loss feedback in order to make suggestions.

While sophisticated closed loop selection methods have been shown to empirically identify good hyperparameter configurations faster (i.e., with fewer iterations) than open loop methods like random search, **two trends have rekindled interest in embarrassingly parallel open loop methods**: 1) modern deep learning models can take days or weeks to train with no signs of efficiency breakthroughs, and 2) the rise of cloud resources available to anyone that charge not by the number of machines, but by the number of CPU-hours used so that 10 machines for 100 hours costs the same as 1000 machines for 1 hour.

This paper explores the landscape of open loop methods, identifying tradeoffs that are rarely considered, if at all acknowledged. While random search is arguably the most popular open loop method and chooses each $x_k$ independently of $\{x_i\}_{i=1}^{k-1}$, it is by no means the only choice. In many ways uniform random search is the least interesting of the methods we will discuss because we will advocate for methods where $x_k$ depends on $\{x_i\}_{i=1}^{k-1}$ to promote **diversity**. In particular, we will focus on

drawing $\{x_i\}_{i=1}^k$ from a $k$-**determinantal point process (DPP)** (Kulesza et al., 2012). DPPs support real, integer, and categorical dimensions—any of which may have a tree structure—and have computationally efficient methods of drawing samples.

Experimentally, we explore the use of our diversity-promoting open-loop hyperparameter optimization method based on $k$-DPP random search. We find that it significantly outperforms uniform random search in cases where the hyperparameter values have a large effect on performance.

Open source implementations of both our hyperparameter optimization algorithm (as an extension to the hyperopt package (Bergstra et al., 2013)) and the MCMC algorithm introduced in Algorithm 2 will be released upon publication.

## 2 RELATED WORK

While this work focuses on open loop methods, the vast majority of recent work on hyperparameter tuning has been on closed loop methods, which we briefly review.

### 2.1 CLOSED LOOP METHODS

Much attention has been paid to sequential model-based optimization techniques such as Bayesian optimization (Snoek et al., 2012; Bergstra et al., 2011b), which sample hyperparameter spaces adaptively. These techniques first choose a point in the space of hyperparameters, then train and evaluate a model with the hyperparameter values represented by that point, then sample another point based on how well previous point(s) performed. These methods can become complicated, and while they can lead to improved performance, the differences are frequently small. In addition, it has recently been observed that many Bayesian optimization methods, when run for $k$ iterations, are outperformed by sampling $2k$ points uniformly at random (Li et al., 2017). Parallelizing Bayesian optimization methods has proven to be nontrivial, and while a number of algorithms exist which sample more than one point at each iteration (Contal et al., 2013; Desautels et al., 2014; González et al., 2016), none can achieve the parallelization that grid search, sampling uniformly, or sampling according to a DPP allow.

One recent line of research has examined the use of DPPs for optimizing hyperparameters, in the context of parallelizing Bayesian optimization (Kathuria et al., 2016; Wang et al., 2017). At each iteration within one trial of Bayesian optimization, instead of drawing a single new point to evaluate from the posterior, they define a DPP over a small region of the space and sample a set of diverse points. While this can lead to easy parallelization *within one iteration* of Bayesian optimization, the overall algorithms are still sequential. Additionally, their approach requires discretizing the hyperparameter space, a drawback which we circumvent.

So-called configuration evaluation methods have been shown to perform well by adaptively allocating resources to different hyperparameter settings (Swersky et al., 2014; Li et al., 2017). They initially choose a set of hyperparameters to evaluate (often uniformly), then partially train a set of models for these hyperparameters. After some fixed training budget (e.g. time, or number of training examples observed), they compare the partially trained models against one another and allocate more resources to those which perform best. Eventually, these algorithms produce one (or a small number) of fully trained, high-quality models. In some sense, these approaches are orthogonal to open vs. closed loop methods since both can be applied with these methods.

### 2.2 OPEN LOOP METHODS

As discussed above, recent trends have renewed interest in open loop methods. And recently, random search was shown to be competitive with sophisticated closed loop methods for modern hyperparameter optimization tasks like deep networks (Li et al., 2017), inspiring other works to explain the phenomenon (Ahmed et al., 2016). Bergstra & Bengio (2012) offer one of the most comprehensive studies of open loop methods to date, and focus attention on comparing random search and grid search. A main takeaway of the paper is that uniform random sampling is generally preferred to

grid search[1] due to the frequent observation that some hyperparameters have little impact on performance, and random search promotes more diversity in the dimensions that matter. Essentially, if points are drawn uniformly at random in $d$ dimensions but only $d' < d$ dimensions are relevant, those same points are uniformly distributed (and just as diverse) in $d'$ dimensions. Grid search, on the other hand, distributes configurations aligned with the axes so if only $d' < d$ dimensions are relevant, many configurations are essentially duplicates.

However, grid search does have one favorable property that is clear in just one dimension. If $k$ points are distributed on $[0, 1]$ on a grid, the maximum spacing between points is equal to $\frac{1}{k-1}$. But if points are uniformly at random drawn on $[0, 1]$, the expected largest gap between points scales as $\frac{1}{\sqrt{k}}$. If you are unlucky enough to have your minimum located in this largest gap, this difference could be considerable. The phenomenon generalizes to higher dimensions but grid search's advantage does not for the reasons above. This is an important concept in numerical integration and one way to quantify this property of a sequence $\mathbf{x} = (x_1, x_2, \ldots, x_k)$ is known as star discrepancy:

$$D_k(\mathbf{x}) = \sup_{u_1, \ldots, u_d \in [0,1]} \left| \frac{1}{k} \sum_{i=1}^{k} \mathbf{1} \left\{ x_i \in \prod_{j=1}^{d} [0, u_j) \right\} - \prod_{j=1}^{d} u_j \right| \tag{1}$$

One can interpret the star discrepancy as a multidimensional version of the Kolmogorov-Smirnov statistic between the sequence $\mathbf{x}$ and the uniform measure. It is well-known that a sequence chosen uniformly at random from $[0, 1]^d$ has an expected star discrepancy of at least $\sqrt{\frac{1}{k}}$ (and is no greater than $\sqrt{\frac{d \log(d)}{k}}$) (Devroye et al., 2013, Corollary 12.5) whereas sequences are known to exist with star discrepancy less than $\frac{\log(k)^d}{k}$ Sobol' (1967), where both bounds depend on absolute constants. These low-discrepancy sequences, as they are known, include the Sobol sequence, which was also given brief mention in (Bergstra & Bengio, 2012) and shown to outperform random search and grid search. We also note that the Sobol sequence is also used as an initialization procedure for some Bayesian Optimization schemes Snoek et al. (2012). However, the Sobol sequence is only defined for continuous spaces, so for hyperparameter search which involves discrete dimensions it is not appropriate.

The final open loop method we study is the DPP, which has been given considerably less attention in the hyperparameter optimization literature. Comparing the star discrepancy of uniform at random and Sobol, one observes that as $d$ grows large relative to $k$, Sobol starts to suffer. Indeed, Bardenet & Hardy (2016) notes that the Sobol rate is not even valid until $k = \Omega(2^d)$ which motivates them to study a formulation of a DPP that has a star discrepancy between Sobol and random and holds for all $k$, small and large. They primarily approached this problem from a theoretical perspective, and didn't include experimental results. Their work, in part, motivates us to look at DPPs as a solution for hyperparameter optimization.

## 3 COMPARISON OF OPEN LOOP METHODS

Optimization performance–how close a point in our sequence is to the true, fixed minimum–is our goal, not a sequence with low discrepancy. However, as Bergstra & Bengio (2012) observed, the rare "large gap" that can occur in random sequences without the low discrepancy property can affect optimization performance, on average. One natural surrogate of average optimization performance is to define a hyperparameter space on $[0, 1]^d$ and measure the distance from a fixed point, say $\frac{1}{2}\mathbf{1} = (\frac{1}{2}, \ldots, \frac{1}{2})$, to the nearest point in the length $k$ sequence in the Euclidean norm squared: $\min_{i=1,\ldots,k} ||x_i - \frac{1}{2}\mathbf{1}||_2^2$. The Euclidean norm (squared) is motivated by a quadratic Taylor series approximation around the minimum of the hypothetical function we wish to minimize. The first question we wish to answer is: is low discrepancy a surrogate for optimization performance? In the first and second columns of Figure 1 we plot the star discrepancy and smallest distance from the center $\frac{1}{2}\mathbf{1}$, respectively, as a function of the length of the sequence, with each row representing dimensions d=2,3,4, for the Sobol sequence, uniform at random, and a DPP (see the next section for details). We

---

[1]Grid search uniformly grids $[0, 1]^d$ such that $x_k = (\frac{i_1}{m}, \frac{i_2}{m}, \ldots, \frac{i_d}{m})$ is a point on the grid for $i_j = 0, 1, \ldots, m$ for all $j$, with a total number of grid points equal to $(m + 1)^d$.

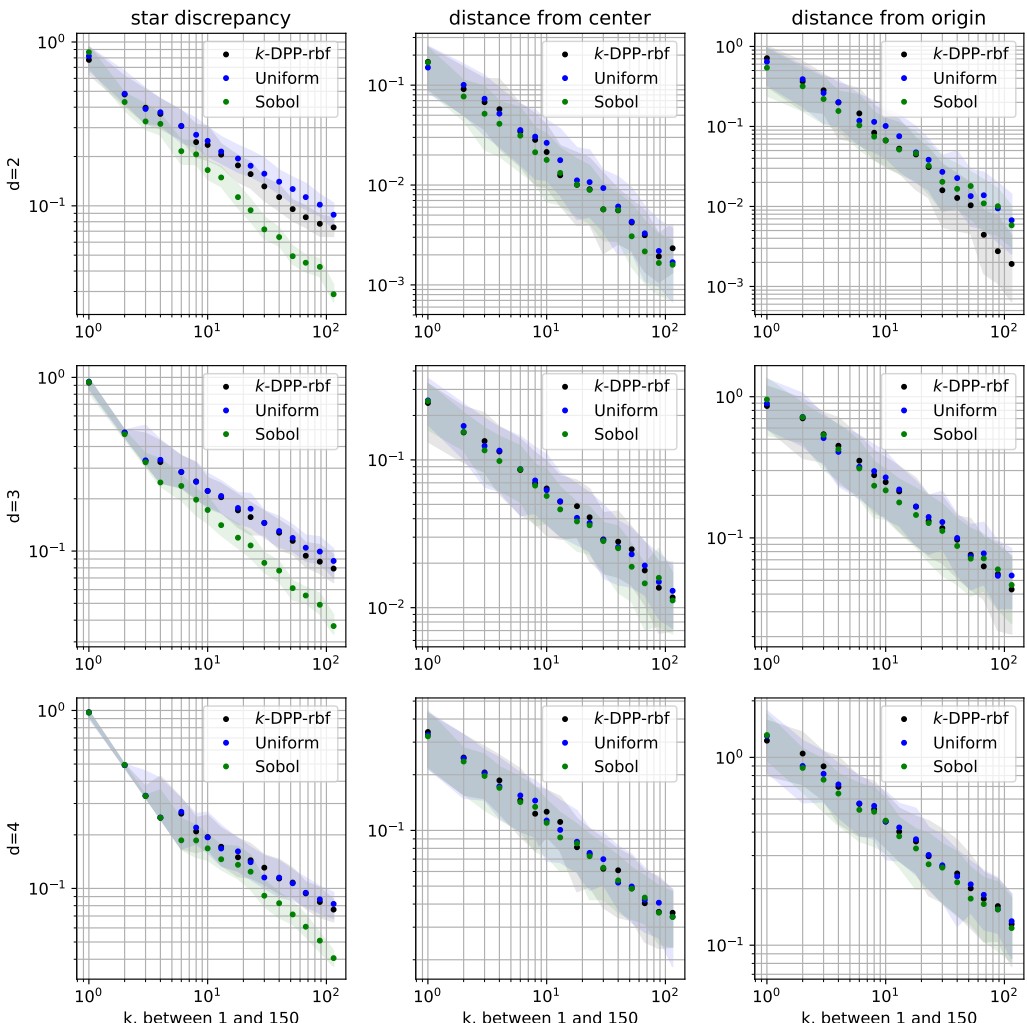

Figure 1: Comparison of the Sobol sequence (with uniform noise), samples a from $k$-DPP, and uniform random for three metrics of interest.

observe that the Sobol sequence is clearly superior in terms of star discrepancy, with the DPP having a slight edge over Uniform. However, all methods appear comparable when it comes to distance to the center.

Acknowledging the fact that practitioners define the search space themselves more often than not, we realize that if the search space bounds are too small, the optimal solution often is found on the edge, or in a corner of the hypercube. Thus, in some situations it makes sense to *bias* the sequence towards the edges and the corners, the very opposite of what low discrepancy sequences attempt to do. While Sobol and uniformly random sequences will not bias themselves towards the corners, a DPP does. This happens because points from a DPP are sampled according to how distant they are from the existing points; this tends to favor points in the corners. This same behavior of sampling in the corners is also very common for Bayesian optimization schemes, which is not surprise due to the known connections between sampling from a DPP and gaussian process (see Section 4.5). In the third column of Figure 1 we plot the distance to the origin which is just an arbitrarily chosen corner of hypercube. As expected, we observe that the DPP tends to outperform uniform at random and Sobol in this metric. In what follows, we study the DPP in more depth and how it performs on real-world hyperparameter tuning problems.

## 4 METHOD

We begin by reviewing determinantal point processes (DPPs) and $k$-DPPs.

Let $\mathcal{B}$ be a domain of values from which we would like to sample a finite subset. (In our use of DPPs, this is the set of hyperparameter settings.) In general, $\mathcal{B}$ could be discrete or continuous; here we assume it is discrete with $N$ values, and we define $\mathcal{Y} = \{1, \ldots, N\}$ to be a a set which indexes $\mathcal{B}$ (this will be particularly useful in Algorithm 1). In Section 4.2 we address when $\mathcal{B}$ has continuous dimensions. A DPP defines a probability distribution over $2^{\mathcal{Y}}$ (all subsets of $\mathcal{Y}$) with the property that two elements of $\mathcal{Y}$ are more (less) likely to both be chosen the more dissimilar (similar) they are. Let random variable $Y$ range over finite subsets of $\mathcal{Y}$.

There are several ways to define the parameters of a DPP. We focus on $\mathbf{L}$-ensembles, which define the probability that a specific subset is drawn (i.e., $P(Y = \mathcal{A})$ for some $\mathcal{A} \subset \mathcal{Y}$) as:

$$P(Y = \mathcal{A}) = \frac{\det(\mathbf{L}_{\mathcal{A}})}{\det(\mathbf{L} + I)}. \tag{2}$$

As shown in Kulesza et al. (2012), this definition of $\mathbf{L}$ admits a decomposition to terms representing the *quality* and *diversity* of the elements of $\mathcal{Y}$. For any $y_i, y_j \in \mathcal{Y}$, let:

$$\mathbf{L}_{i,j} = q_i q_j \mathcal{K}(\phi_i, \phi_j), \tag{3}$$

where $q_i > 0$ is the quality of $y_i$, $\phi_i \in R^d$ is a featurized representation of $y_i$, and $\mathcal{K} : R^d \times R^d \to [0, 1]$ is a similarity kernel (e.g. cosine distance). (We will discuss how to featurize hyperparameter settings in Section 4.3.)

Here, we fix all $q_i = 1$; in future work, closed loop methods might make use of $q_i$ to encode evidence about the quality of particular hyperparameter settings to adapt the DPP's distribution over time.

### 4.1 SAMPLING FROM A $k$-DPP

DPPs have support over all subsets of $\mathcal{Y}$, including $\emptyset$ and $\mathcal{Y}$ itself. In many practical settings, one may have a fixed budget that allows running the training algorithm $k$ times, so we require precisely $k$ elements of $\mathcal{Y}$ for evaluation. $k$-DPPs are distributions over subsets of $\mathcal{Y}$ of size $k$. Thus,

$$P(Y = \mathcal{A} \mid |Y| = k) = \frac{\det(\mathbf{L}_{\mathcal{A}})}{\sum_{\mathcal{A}' \subset \mathcal{Y}, |\mathcal{A}'| = k} \det(\mathbf{L}_{\mathcal{A}'})}. \tag{4}$$

### 4.2 NEW MCMC ALGORITHM

Kulesza et al. (2012) give an algorithm for sampling exactly from $k$-DPPs, though it runs in $O(N^3)$; a Metropolis-Hastings algorithm presented by Anari et al. (2016) is a simple and fast alternative (included here as Algorithm 1). Both of these sampling algorithms assume the DPP is defined over a finite number of items; they are restricted to discrete domains. We propose a generalization of the MCMC algorithm which preserves relevant computations while allowing sampling from base sets with discrete dimensions, continuous dimensions, or some continuous and some discrete dimensions (Algorithm 2). To the best of our knowledge, this is the first algorithm which allows for sampling from a $k$-DPP defined over mixed discrete and continuous spaces.

Algorithm 1 proceeds as follows: First, initialize a set $Y$ with $k$ indices of $\mathbf{L}$, drawn uniformly. Then, at each iteration, sample two indices of $\mathbf{L}$ (one within and one outside of the set $Y$), and with some probability replace the item in $Y$ with the other.

When we have continuous dimensions in the base set, however, we can't define the matrix $\mathbf{L}$, so sampling indices from it is not possible. We propose Algorithm 2, which samples points directly from the base set $\mathcal{B}$ instead (assuming continuous dimensions are bounded), and computes only the principal minors of $\mathbf{L}$ needed for the relevant computations on the fly.

Even in the case where the dimensions of $\mathcal{B}$ are discrete, Algorithm 2 requires less computation and space than Algorithm 1 (assuming the quality and similarity scores are stored once computed, and retrieved when needed). Previous analyses claimed that Algorithm 1 should be run for $O(N \log(N))$

---

**Algorithm 1** Drawing a sample from a discrete $k$-DPP

---

**Input:** $\mathbf{L}$, a symmetric, $N \times N$ matrix where $\mathbf{L}_{i,j} = q_i q_j \mathcal{K}(\phi_i, \phi_j)$ which defines a DPP over a finite base set of items $\mathcal{B}$, and $\mathcal{Y} = \{1, \ldots, N\}$, where $\mathcal{Y}_i$ indexes a row or column of $\mathbf{L}$
**Output:** $\mathcal{B}_{\mathbf{Y}}$ (the points in $\mathcal{B}$ indexed by $\mathbf{Y}$)
  1: Initialize $\mathbf{Y}$ to $k$ elements sampled from $\mathcal{Y}$ uniformly
  2: **while** not mixed **do**
  3:     uniformly sample $u \in \mathbf{Y}, v \in \mathcal{Y} \setminus \mathbf{Y}$
  4:     set $\mathbf{Y}' = \mathbf{Y} \cup \{v\} \setminus \{u\}$
  5:     $p \leftarrow \frac{1}{2} min(1, \frac{\det(\mathbf{L}_{\mathbf{Y}'})}{\det(\mathbf{L}_{\mathbf{Y}})})$
  6:     with probability $p$: $\mathbf{Y} = \mathbf{Y}'$
  7: Return $\mathcal{B}_{\mathbf{Y}}$

---

**Algorithm 2** Drawing a sample from a $k$-DPP defined over a space with continuous and discrete dimensions

---

**Input:** A base set $\mathcal{B}$ with some continuous and some discrete dimensions, a quality function $\mathbf{\Psi}$ : $\mathbf{Y}_i \rightarrow q_i$, a feature function $\mathbf{\Phi} : \mathbf{Y}_i \rightarrow \phi_i$
**Output:** $\boldsymbol{\beta}$, a set of $k$ points in $\mathcal{B}$
  1: Initialize $\boldsymbol{\beta}$ to $k$ points sampled from $\mathcal{B}$ uniformly
  2: **while** not mixed **do**
  3:     uniformly sample $u \in \boldsymbol{\beta}, v \in \mathcal{B} \setminus \boldsymbol{\beta}$
  4:     set $\boldsymbol{\beta}' = \boldsymbol{\beta} \cup \{v\} \setminus \{u\}$
  5:     compute the quality score for each item, $q_i = \mathbf{\Psi}(\boldsymbol{\beta}_i), \forall i$, and $q_i' = \mathbf{\Psi}(\boldsymbol{\beta}_i'), \forall i$
  6:     construct $\mathbf{L}_{\boldsymbol{\beta}} = [q_i q_j \mathcal{K}(\mathbf{\Phi}(\boldsymbol{\beta}_i), \mathbf{\Phi}(\boldsymbol{\beta}_j))], \forall i, j$
  7:     construct $\mathbf{L}_{\boldsymbol{\beta}'} = [q_i' q_j' \mathcal{K}(\mathbf{\Phi}(\boldsymbol{\beta}_i'), \mathbf{\Phi}(\boldsymbol{\beta}_j'))], \forall i, j$
  8:     $p \leftarrow \frac{1}{2} min(1, \frac{\det(\mathbf{L}_{\boldsymbol{\beta}'})}{\det(\mathbf{L}_{\boldsymbol{\beta}})})$
  9:     with probability $p$: $\boldsymbol{\beta} = \boldsymbol{\beta}'$
 10: Return $\boldsymbol{\beta}$

---

steps. There are $O(N^2)$ computations required to compute the full matrix $L$, and at each iteration we will compute at most $O(k)$ new elements of $L$, so even in the worst case we will save space and computation whenever $k \log(N) < N$. In expectation, we will save significantly more.

### 4.3 Constructing $\mathbf{L}$ for hyperparameter optimization

The vector $\phi_i$ will encode $y_i$ (an element of $\mathcal{Y}$), which in its most general form is an attribute-value mapping assigning values to different hyperparameters.

Let $\phi_i$ be a feature vector for $y_i \in \mathcal{Y}$, a modular encoding of the attribute-value mapping, in which fixed segments of the vector are assigned to each hyperparameter attribute (e.g., the dropout rate, the choice of nonlinearity, etc.). For a hyperparameter that takes a numerical value in range $[h_{\min}, h_{\max}]$, we encode value $h$ using one dimension ($j$) of $\phi$ and project into the range $[0, 1]$:

$$\phi[j] = \frac{h - h_{\min}}{h_{\max} - h_{\min}} \tag{5}$$

This rescaling prevents hyperparameters with greater dynamic range from dominating the similarity calculations. A categorical-valued hyperparameter attribute that takes $m$ values is given $m$ elements of $\mathbf{r}$ and a one-hot encoding. We then compute similarity using an RBF kernel, $\mathcal{K} = \exp\left(-\frac{||\phi_i - \phi_j||^2}{2\sigma^2}\right)$, and hence label our approach $k$-DPP-RBF. Values for $\sigma^2$ lead to models with different properties; when $\sigma^2$ is small, points that are spread out have little impact, and when $\sigma^2$ is large, the increased repulsion between the points encourages them to be as far apart as possible. This tradeoff is represented in Figure 1.

### 4.4 Tree-structured hyperparameters

Many real-world hyperparameter search spaces are tree-structured. For example, the number of layers in a neural network is a hyperparameter, and each additional layer adds at least one new hyperparameter which ought to be tuned (the number of nodes in that layer). For a binary hyperparameter like whether or not to use regularization, we use a one-hot encoding. When this hyperparameter is "on," we set the associated regularization strength as above, and when it is "off" we set it to zero. Intuitively, with all other hyperparameter settings equal, this causes the off-setting to be closest to the least strong regularization. One can also treat higher-level design decisions as hyperparameters (Komer et al., 2014), such as whether to train a logistic regression classifier, a convolutional neural network, or a recurrent neural network. In this construction, the type of model would be a categorical variable (and thus get a one-hot encoding), and all child hyperparameters for an "off" model setting (such as the convergence tolerance for logistic regression, when training a recurrent neural network) would be set to zero.

### 4.5 Connection to Gaussian processes

Gaussian processes are used widely in hyperparameter optimization algorithms. Hennig & Garnett (2016) claim that sampling from a DPP with kernel $\mathcal{K}$ is equivalent to sequentially sampling proportional to the posterior variance of a GP defined with covariance kernel $\mathcal{K}$. Since the entropy of a Gaussian is proportional to the log determinant of the covariance matrix, points drawn from a DPP have probability proportional to $\exp(\text{information gain})$, and the most probable set from the DPP is the set which maximizes the information gain.

## 5 Hyperparameter Optimization Experiments

In this section we present our hyperparameter optimization experiments. We compare $k$-DPP-RBF, uniform sampling, and a Bayesian optimization algorithm in Section 5.1. We compare samples drawn using Algorithm 1 (which necessitates discretizing the hyperparameter space) and Algorithm 2 against samples drawn uniformly at random in Section 5.2. It is worth noting that as $k$ increases, all sampling methods approach the true optimum.

### 5.1 Convolutional neural networks for text classification

Our experiments consider a setting where hyperparameters have a large effect on performance: a convolutional neural network for text classification (Kim, 2014). The task is binary sentiment analysis on the Stanford sentiment treebank (Socher et al., 2013). On this balanced dataset, random guessing leads to 50% accuracy. We use the CNN-non-static model from Kim (2014), with word2vec (Mikolov et al., 2013) vectors. The model architecture consists of a convolutional layer, a max-over-time pooling layer, then a fully connected layer leading to a softmax.

We begin with a search over three hyperparameters, assuming a budget of $k = 20$ repetitions of training the convolutional neural net. $L_2$ regularization strengths in the range $[e^{-5}, e^{-1}]$ (or no regularization) and dropout rates in $[0.0, 0.7]$ are considered. We consider three increasingly "easy" ranges for the learning rate:

- Hard: $[e^{-5}, e^5]$, where the majority of the range leads to accuracy no better than chance.
- Medium: $[e^{-5}, e^{-1}]$, where half of the range leads to accuracy no better than chance.
- Easy: $[e^{-10}, e^{-3}]$, where the entire range leads to models that beat chance.

Figure 2 shows the accuracy (averaged over 50 runs) of the best model found after exploring 1, 2, . . . , $k$ hyperparameter settings. We see that $k$-DPP-RBF finds better models with fewer iterations necessary than the other approaches, especially in the most difficult case. Figure 2 compares the sampling methods against a Bayesian optimization technique using a tree-structured Parzen estimator (BO-TPE; Bergstra et al., 2011b). This technique evaluates points sequentially, allowing the model to choose the next point based on how well previous points performed (a closed loop approach). It is state-of-the-art on tree-structured search spaces (though its sequential nature limits parallelization). Surprisingly, we find it performs the worst, even though it takes advantage of additional information.

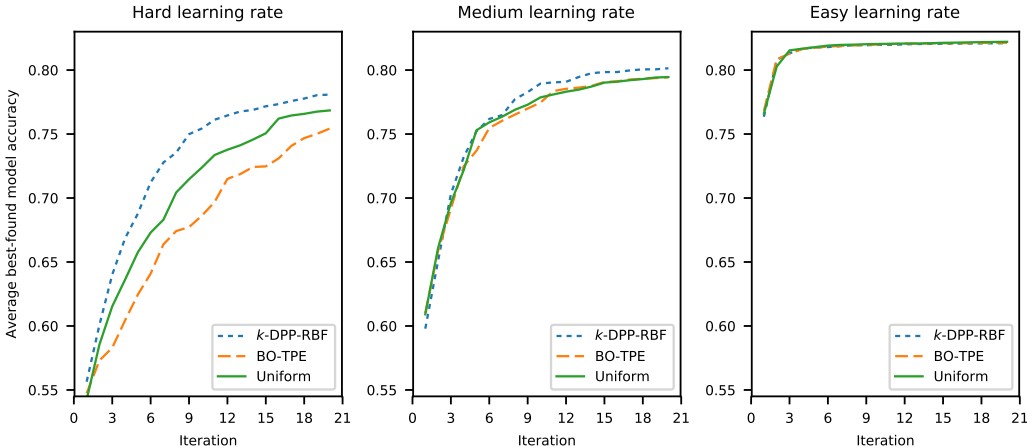

Figure 2: Average best-found model accuracy by iteration when training a convolutional neural network on three hyperparameter search spaces (defined in Section 5.1), averaged across 50 trials of hyperparameter optimization, with $k = 20$.

We hypothesize that the exploration/exploitation tradeoff in BO-TPE causes it to commit to more local search before exploring the space fully, thus not finding hard-to-reach global optima.

Note that when considering points sampled uniformly or from a DPP, the order of the $k$ hyperparameter settings in one trial is arbitrary (though this is not the case with BO-TPE as it is an iterative algorithm). The variance of the $k$-DPP methods (not shown for clarity) tends to be high in early iterations, simply because the $k$ samples from a $k$-DPP are likely to be more diverse than those sampled uniformly, but in all cases the variance of the best of the $k$ points is lower than when sampled uniformly.

## 5.2 OPTIMIZING WITHIN RANGES KNOWN TO BE GOOD

Zhang & Wallace (2015) analyzed the stability of convolutional neural networks for sentence classification with respect to a large set of hyperparameters, and found a set of six which they claimed had the largest impact: the number of kernels, the difference in size between the kernels, the size of each kernel, dropout, regularization strength, and the number of filters. We optimized over their prescribed "Stable" ranges; average accuracies across 50 trials of hyperparameter optimization are shown in Figure 3, across $k = 20$ iterations, with each dimension discretized to five values (for the discretized experiments). For both uniform sampling and sampling using $k$-DPP-RBF, discretizing the search space hurts performance, thus motivating the use of Algorithm 2. Additionally, we find that even in this case where every value gives reasonable performance, $k$-DPP-RBF sampling outperforms uniform sampling.

Our experiments reveal that, while the hyperparameters proposed by Zhang & Wallace (2015), can have an effect, the learning rate, which they don't analyze, is at least as impactful.

## 6 CONCLUSIONS

We have explored open loop hyperparameter optimization built on sampling from $k$-DPPs. We described how to construct $k$-DPPs over hyperparameter search spaces, and showed that sampling from these retains the attractive parallelization capabilities of random search. Our experiments demonstrate that, under a limited computation budget, on a number of realistic hyperparameter optimization problems, these approaches perform better than sampling uniformly at random. As we increase the difficulty of our hyperparameter optimization problem (i.e., as values which lead to good model

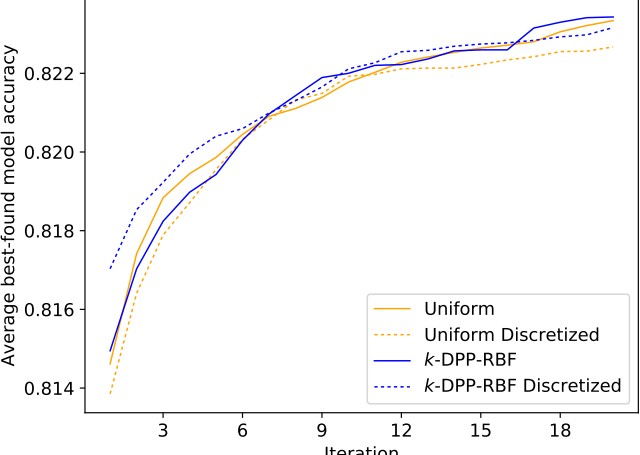

Figure 3: Average best-found model accuracy by iteration when training a convolutional neural network on the "Stable" search space (defined in Section 5.2), averaged across 50 trials of hyperparameter optimization, with $k = 20$. Discretizing the space reduces the accuracy found for both uniform sampling and $k$-DPP-RBF, but in both cases $k$-DPP-RBF finds better optima than uniform sampling.

evaluations become more scarce) the improvement over sampling uniformly at random increases. An open-source implementation of our method is available.[2]

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
