# OpenReview forum: "Open Loop Hyperparameter Optimization and Determinantal Point Processes"
_ICLR.cc/2018/Conference — Reject_

### Official Review · AnonReviewer3 · 2017-11-26
**Limited Scope, Weak Evaluation**

**Rating:** 4
**Confidence:** 5

**Review:**


This paper considers hyperparameter searches in which all of the
candidate points are selected in advance.  The most common approaches
are uniform random search and grid search, but more recently
low-discrepancy sequences have sometimes been used to try to achieve
better coverage of the space.  This paper proposes using a variant of
the determinantal point process, the k-DPP to select these points.
The idea is that the DPP provides an alternative form of diversity to
low-discrepancy sequences.

Some issues I have with this paper:

1. Why a DPP? It's pretty heavyweight. Why not use any of the other
(potentially cheaper) repulsive point processes that also achieve
diversity?  Is there anything special about it that justifies this
work?

2. What about all of the literature on space-filling designs, e.g.,
latin hypercube designs?  Statisticians have thought about this for a
long time.

3. The motivation for not using low-discrepancy sequences was discrete
hyperparameters.  In practice, people just chop up the space or round.
Is a simple kernel with one length scale on a one-hot coding adding
value? In this setup, each parameter can only contribute "same or
different" to the diversity assessment.  In any case, the evaluations
didn't have any discrete parameters.  Given that the discrete setting
was the motivation for the DPP over LDS, it seems strange not to even
look at that case.

4. How do you propose handling ordinal variables? They're a common
case of discrete variables but it wouldn't be sensible to use a
one-hot coding.

5. Why no low discrepancy sequence in the experimental evaluation of
section 5?  Since there's no discrete parameters, I don't see what the
limitation is.

6. Why not evaluate any other low discrepancy sequences than Sobol?

7. I didn't understand the novelty of the MCMC method relative to
vanilla M-H updates.  It seems out of place.

8. The figures really need error bars --- Figure 3 in particular.  Are
these differences statistically significant?

---

> ### Author Response · Authors · 2018-01-05
> **Re: Limited Scope, Weak Evaluation**
>
> We thank AnonReviewer3 for their response. We address the points in order.
>
> 1: We show how to efficiently draw samples from arbitrary tree-structured hyperparameter spaces which include continuous and discrete dimensions, which is not clear for other point processes. We also highlight connections between our approach and GP-based BO, which is the most common approach used currently.
>
> 2: Bergstra and Bengio, 2012 (http://www.jmlr.org/papers/volume13/bergstra12a/bergstra12a.pdf) found that the Sobol sequence outperformed latin hypercube sampling, and that the Niederreiter and Halton sequences were similar to Sobol. We will add this point to our paper.
>
> 3a: The primary motivation for not using a low-discrepancy sequence is actually the tree-structured hyperparameters. It isn't immediately obvious how to map from a sequence in [0,1]^d to arbitrary tree structures.
>
> 3b: The experiments in section 5.1 include a categorical hyperparameter (whether or not to regularize), and the experiments in section 5.2 include a number of discrete hyperparameters (e.g. the number of kernels). We will make this more clear in the paper.
>
> 4: Great question. As our approach doesn't get any information about the function values (i.e. we're not learning the kernel), we propose to use a unary encoding which evenly spaces the values for ordinal variables. A three-value ordinal variable would then be [1,0,0], [1,1,0], [1,1,1]. If we have any a priori knowledge about how far apart the ordinal variables are, it's simple to include that in this encoding.
>
> 5: see response 3c.
>
> 6: see response 2.
>
> 7: The novelty in Algorithm 2 is in the proposal distribution: instead of sampling uniformly from a discrete set of items (as in Algorithm 1), we draw samples directly from the space over which the DPP is defined (which potentially has continuous and discrete dimensions). It is correct that it uses M-H updates. We can make the novelty more explicit.
>
> 8: All experimental results are statistically significant (when k > ~15). For example, in figure 3, k-DPP-RBF has a 99% confidence interval of [82.342,82.344], while uniform has a 99% confidence interval of [82.266,82.268]. We excluded confidence intervals only for readability, but will add this to the text of the paper.

---

### Official Review · AnonReviewer1 · 2017-11-27
**comparison with recent work and scalability**

**Rating:** 4
**Confidence:** 5

**Review:**

The authors propose k-DPP as an open loop (oblivious to the evaluation of configurations) method for hyperparameter optimization and provide its empirical study and comparison with other methods such as grid search, uniform random search, low-discrepancy Sobol sequences, BO-TPE (Bayesian optimization using tree-structured Parzen estimator) by Bergstra et al. (2011). The k-DPP sampling algorithm and the concept of k-DPP-RBF over hyperparameters are not new, so the main contribution here is the empirical study.

The first experiment by the authors shows that k-DPP-RBF gives better star discrepancy than uniform random search while being comparable to low-discrepancy Sobol sequences in other metrics such as distance from the center or an arbitrary corner (Fig. 1).

The second experiment shows surprisingly that for the hard learning rate range, k-DPP-RBF performs better than uniform random search, and moreover, both of these outperform BO-TPE (Fig. 2, column 1).

The third experiment shows that on good or stable ranges, k-DPP-RBF and its discrete analog slightly outperform uniform random search and its discrete analog, respectively.

I have a few reservations. First, I do not find these outcomes very surprising or informative, except for the second experiment (Fig. 2, column 1). Second, their study only applies to a small number like 3-6 hyperparameters with a small k=20. The real challenge lies in scaling up to many hyperparameters or even k-DPP sampling for larger k. Third, the authors do not compare against some relevant, recent work, e.g., Springenberg et al. (http://aad.informatik.uni-freiburg.de/papers/16-NIPS-BOHamiANN.pdf) and Snoek et al. (https://arxiv.org/pdf/1502.05700.pdf) that is essential for this kind of empirical study.

---

> ### Author Response · Authors · 2018-01-05
> **Re: comparison with recent work and scalability**
>
> We thank AnonReviewer1 for their response. We will address their points in order.
>
> - We chose our experiments to represent common machine learning setups, but sampling from a k-DPP extends well into higher dimensions and larger k. This can be seen through the connection to GPs (section 4.5): one can generate a draw from a k-DPP by sequentially sampling from (and updating) the posterior variance of a GP. This scales with O(d*k^3), where d is the dimension. A quick experiment shows drawing samples up to k=500 takes less than twenty minutes in small dimensions on a c4.8xlarge AWS EC2 instance (which has 36 cores). As noted in the paper, as k increases, all methods approach the true max, so differences between methods are primarily found with smaller k.
>
> - We thank the reviewer for the suggestions for further comparisons. Both given citations use the same parallelization approach (from Snoek et al., 2012, equation 7). In the open loop case, their approach (which approximates the posterior using MCMC then uses their acquisition function to choose the next point) reduces to a variant of k-DPP-RBF that has less repulsive properties. We will add this to our experiment section.

---

### Official Review · AnonReviewer2 · 2017-11-27
**An interesting idea for pure exploration hyperparameter tuning, but needs to be compared to more recent existing methods.**

**Rating:** 4
**Confidence:** 5

**Review:**

In this paper, the authors consider non-sequential (in the sense that many hyperparameter evaluations are done simultaneously) and uninformed (in the sense that the hyperparameter evaluations are chosen independent of the validation errors observed) hyperparameter search using determinantal point processes (DPPs). DPPs are probability distributions over subsets of a ground set with the property that subsets with more "diverse" elements have higher probability. Diverse here is defined using some similarity metric, often a kernel. Under the RBF kernel, the more diverse a set of vectors is, the closer the kernel matrix becomes to the identity matrix, and thus the larger the determinant (and therefore probability under the DPP) grows. The authors propose to do hyperparameter tuning by sampling a set of hyperparameter evaluations from a DPP with the RBF kernel.

Overall, I have a couple of concerns about novelty as well as the experimental evaluation for the authors to address. As the authors rightly point out, sampling hyperparameter values from a DPP is equivalent to sampling proportional to the posterior uncertainy of a Gaussian process, effectively leading to a pure exploration algorithm. As the authors additionally point out, such methods have been considered before, including methods that directly propose to batch Bayesian optimization by choosing a single exploitative point and sampling the remainder of the batch from a DPP (e.g., [Kathuria et al., 2016]). The default procedure for parallel BayesOpt used by SMAC [R2] is (I believe) also to choose a purely explorative batch. I am unconvinced by the argument that "while this can lead to easy parallelization within one iteration of Bayesian optimization, the overall algorithms are still sequential." These methods can typically be expanded to arbitrarily large batches and fully utilize all parallel hardware. Most implementations of batch Bayesian optimization in practice (SMAC and Spearmint as examples) will even start new jobs immediately as jobs finish -- these implementations do not wait for the entire batch to finish typically.

Additionally, while there has been some work extending GP-based BayesOpt to tree-based parameters [R3], at a minimum SMAC in particular is known well suited to the tree-based parameter search considered by the authors. I am not sure that I agree that TPE is state-of-the-art on these problems: SMAC typically does much better in my experience.

Ultimately, my concern is that--considering these tools are open source and relatively stable software at this point--if DPP-only based hyperparameter optimization is truly better than the parallelization approach of SMAC, it should be straightforward enough to download SMAC and demonstrate this. If the argument that BayesOpt is somehow "still sequential" is true, then k-DPP-RBF should outperform these tools in terms of wall clock time to perform optimization, correct?

[R1] Kathuria, Tarun and Deshpande, Amit and Kohli, Pushmeet. Batched Gaussian Process Bandit Optimization via Determinantal Point Processes, 2016.

[R2] Several papers, see: http://www.cs.ubc.ca/labs/beta/Projects/SMAC/

[R3] Jenatton, R., Archambeau, C., González, J. and Seeger, M., 2017, July. Bayesian Optimization with Tree-structured Dependencies. In International Conference on Machine Learning (pp. 1655-1664).

---

> ### Author Response · Authors · 2018-01-05
> **Re: An interesting idea for pure exploration hyperparameter tuning, but needs to be compared to more recent existing methods.**
>
> We thank AnonReviewer2 for their review. We will address their points in order.
>
> - The work of Kathuria et al., 2016 is quite related to our experiment section, but has a few differences. While Kathuria et al., 2016 do sample from a DPP within each batch, they only evaluate their approach as part of a sequential BO algorithm, and it is unclear how much of their improvement is from the DPP and how much is from the acquisition function. Their approach is to first maximize their acquisition function (EST), then draw a batch sample using a k-DPP defined only in a relevance region around that point. They also discretize the space, which our results show leads to worse results. We will describe their work in further detail in our paper, so the differences are more clear.
>
> - Parallel optimization in SMAC is described in Hutter et al., 2012. First, they sample points using latin hypercube sampling (which Bergstra and Bengio 2012 [http://www.jmlr.org/papers/volume13/bergstra12a/bergstra12a.pdf] found was outperformed by the Sobol sequence). Then, at each iteration, they choose k points by maximizing the "optimistic confidence bound" (-mu + lambda * sigma), for a set of k values for lambda independently drawn from an exponential distribution, where mu and sigma are the posterior mean and variance predicted by their decision tree. In the fully parallel setup our work addresses, mu and sigma are not updated, so fully parallel SMAC is equivalent to uniform sampling.
>
> - Our work addresses the fully parallel case (e.g. running experiments on Amazon EC2 instances, where running one instance for ten hours costs the same as running ten instances for one hour). Thus, starting new jobs immediately as other jobs finish is actually inefficient and not desirable -- if we have the budget to run more jobs, we start them all at the same time (before any jobs finish).
>
> - We will run experiments using sequential SMAC in addition to our experiments using TPE, but we remind the reviewer that both of these approaches are using information unavailable to our open-loop methods. We focus on the fully parallel case because we have practically unlimited parallelization hardware (AWS EC2 instances). For a fixed budget of k evaluations, TPE and sequential SMAC take k times longer than open loop methods (using the simplifying assumption that all evaluations take the same amount of time). Even batch methods are at least twice as slow (with the fastest batch method running two iterations of batch size k/2).

---

### Decision · Program_Chairs · 2018-01-29
**ICLR 2018 Conference Acceptance Decision**

**Decision:**

Reject

**Comment:**

The idea of using the determinant of the covariance matrix over inputs to select experiments to run is a foundational concept of experimental design.  Thus it is natural to think about extending such a strategy to sequential model based optimization for the hyperparameters of machine learning models, using recent advances in determinantal point processes.  The idea of sampling from k-DPPs to do parallel hyperparameter search, balancing quality and diversity of expected outcomes, seems neat.  While the reviewers found the idea interesting, they saw weaknesses in the approach and most importantly were not convinced by the empirical results.  All reviewers thought that the baselines were inappropriate given recent work in hyperparameter optimization (and classic work in statistics).

Pros:
- Useful to a large portion of the community (if it works)
- An interesting idea that seems timely

Cons:
- Only slightly outperforms baselines that are too weak
- Not empirically compared to recent literature
- Some of the design and methodology require more justification
- Experiments are limited to small scale problems